# Influences of roughness and filling degree on the shear strength and damage evolution characteristics of cement-filled joints

Zhilin Shu[1], Yang Liu[1]*, Yicheng Ye[1,2], Weiqi Wang[3], Binyu Luo[1,2], Jinpeng Jia[1]

1 School of Resources and Environmental Engineering, Wuhan University of Science and Technology, Wuhan, China, 2 Hubei Key Laboratory for Efficient Utilization and Agglomeration of Metallurgic Mineral Resources, Wuhan, China, 3 Key Laboratory of Ministry of Education on Safe Mining of Deep Metal Mines, Northeastern University, Shenyang, China

* liuyang@wust.edu.cn

**Data Availability Statement:** All relevant data are within the paper.

**Funding:** The authors acknowledge financial support from the Wuhan Knowledge Innovation

## Abstract

The shear resistance of filling joints is an important factor affecting the stability of rock joints. Pressure-shear tests of cement-filled joints were carried out. Combined with the acoustic emission (AE) technique, the effects of normal stress, roughness and filling degree on the shear strength, damage morphology and damage evolution of cement-filled joints were investigated. The results show that with the increase of roughness, the failure mode is more complicated. When the roughness is low, only the bonding surface of the interface between the filler and the joint surface is damaged, and the filling degree has a weak effect on the failure mode. When the roughness is high, with the increase of normal stress and filling degree, the failure of the filled joint is from the joint failure of the bonding surface and the filling material to the serious failure of the bonding surface, the filling material and the joint. The peak shear strength of filled joints is positively correlated with roughness and negatively correlated with filling degree. With the increase of filling degree, the influence of roughness will be weakened by filling material, and the normal stress will amplify the effect of roughness. The evolution characteristics of AE show that the damage degree of filled joints is positively correlated with normal stress and roughness, and negatively correlated with filling degree. The joint surface is damaged locally at first, then failure near the main raised body of the joint, and finally spreads to the whole joint surface.

## 1 Introduction

Rock joints are widespread in nature and they often act as weak surfaces that are prone to sliding [1–3]. Many engineering phenomena have shown [4, 5] that shear slip and damage of discontinuous surfaces not only lead to rock instability, but also induce geologic hazards and cause damage to the ecological environment [4, 6, 7]. Cement filling has been widely used to reinforce rock joints in order to avoid irreversible disasters and to safeguard the stability of rock structures. In order to ensure the best effect of filling reinforcement, it is crucial to

Special Program (20220108010203 07), The 2022
Hubei Emergency Management Department
Special Funds for Safety Production Program
(SJZX20220907), Hubei Provincial Natural Science
Foundation Youth Program(2022CFB590), National
Natural Science Foundation of China (42307237).
The funders had a role in study design, data
collection and analysis, decision to publish, and
preparation of the manuscript.

**Competing interests:** The authors have declared
that no competing interests exist.

accurately evaluate the shear strength of rock joints, so efforts have been made to study the mechanical properties and shear behavior of joints under different conditions [8–11].

A large number of scholars have conducted numerous studies on the shear behavior of unfilled joints, which shows that normal stress and roughness are the main factors affecting the shear strength and failure mode of joints. Wang [12] found that under the action of low and high normal stress, the joint exhibits two failure modes at the mesoscopic level: elastic shear friction, sliding and elastic extrusion wear. The experimental results of He [13] and Yin [14] show that the peak shear strength and peak shear displacement are strongly influenced by roughness, normal phase force and shear rate. The damage pattern increases with the increase of shear rate. Liu [15] and Meng [16] found that the failure types will change with the increase of normal stress, and the failure forms are shear failure under low normal stress and tensile-shear mixed failure under medium and high normal stress. Wu [17] and Xiong [18] found that the morphological characteristics of the shear curve were related to roughness and normal stress, and the curve gradually becomes steep with the increase of roughness. With the increase of normal stress, the climbing effect weakens and the peak shear stress increases. Adrien [19] integrated roughness into the Mohr-Coulomb shear model, which was able to accurately predict the peak shear strength of non-filled joints, and thus assessed the effect of joint roughness on shear strength. The research shows that the shear strength increases obviously with the increase of roughness, but the increase of normal stress will weaken the influence of roughness on the shear strength of contact surface [20, 21].

For filled joints, the filling degree is a representation of the filling degree (Fig 1), the filling degree is expressed as t/d (t is the filling thickness, d is the maximum undulation height of the joint surface), which has a significant influence on the shear mechanical properties of cement-grouted rock joints. There are two critical filling degrees, namely 0.1 and 1.0. When the filling degree is less than 1.0, the shear strength increases rapidly with increasing degree of filling. Between 0.1 and 1.0, the shear strength decreases with increasing filling degree. At greater than 1.0, the shear mechanical parameters are almost constant [22]. The shear stress-shear displacement curve of filled joints includes three shear stages: quasi-elastic stage, failure stage and residual shear stage, and the peak shear strength decreases with the increase of filling degree [22, 23]. Due to the presence of the filling body, the shear failure pattern of the joint will be more complex. The results show that joint roughness has a great influence on the shear failure of filled joints, and the shear failure of joints shows three different forms [24]. The influence of the filling body on the shear stress-strain curve is mainly manifested as a large slope in the

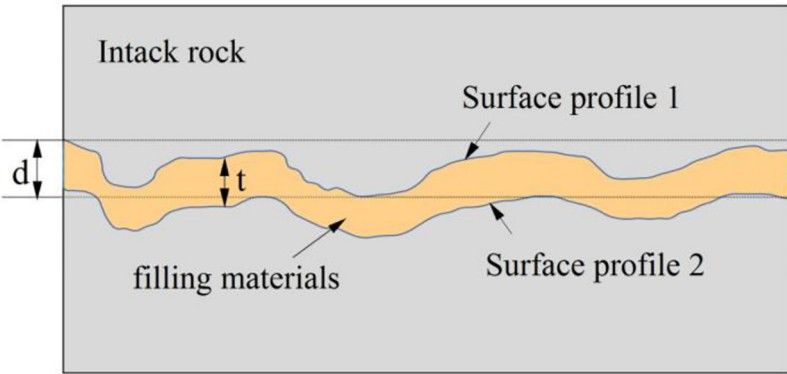

**Fig 1. Filled nodes and filling degree(t/d) diagram.**

elastic stage and a small slope in the plastic stage [15, 25]. Through the shear tests of unfilled joints and filled joints, Tang [26] and Hu [27] found that with the increase of normal stress, the shear stress increases and the normal displacement decreases. Compared with unfilled joints, the influence of roughness of filled joints is usually reduced.

Acoustic emission (AE) technology enables the characterization of damage evolution in specimens through microscopic monitoring [28]. AE technology is simple to operate and requires low samples. The evolution of AE parameters mainly includes the analysis of events, impacts, counts and energies [29, 30]. Shang [31] monitored the rupture process inside the specimen during shear by the AE technique. The AE events were found to be dispersed in the pre-peak phase but confined around the cleavage surface in the post-peak phase. Ostapchuk [32] studied the acoustic emission behavior of filled fissures in different states of shear deformation. The analysis of the evolution of the acoustic emission parameters allows to construct the process of damage evolution inside the specimen. Meng [33] found that for filled joints, the peak event rate and energy rate are mainly obtained at or near the turn of the shear stress curve. The evolution of acoustic emission with displacement can accurately reflect the shear damage mechanism during shearing.

The current status of the above studies shows that normal stress, roughness and filling degree significantly affect the shear behavior of rock joints. However, limited by the specific rock environment and engineering background, the existing research mainly focuses on individual or partial factors. In engineering practice, many factors simultaneously affect the shear characteristics of filled joints. Therefore, shear tests on filled joints under varying normal stresses, roughness and filling degrees are essential. By integrating the acoustic emission (AE) technique, this study investigates the combined effects of multiple factors on the shear strength, damage evolution and damage characteristics of filled joints.

## 2 Filled joint shear test

### 2.1 Sample preparation

Select 425 cement, sand and water, respectively, in accordance with the ratio of 1:0.62:0.4 made of joint specimens [34], with cement paste as filling material, made of different roughness and filling degree (0.5, 1.0 and 1.5, respectively) of the filling joint specimens Based on the 10 standard roughness (JRC) curves proposed by Barton [35] and combined with the reconstruction of standard roughness profiles in the original software using digitized data [36], three typical standard roughness curves were selected for the study (Fig 2), which are 2.8, 10.8 and 18.7 respectively. The undulation height differences of the three curves are 1.7 mm, 5.4 mm, and 4.4 mm, respectively. i.e., the three filling degrees correspond to filling thicknesses of 0.85, 1.7, and 2.55 mm, 2.7, 5.4, and 8.1 mm, and 2.2, 4.4, and 6.6 mm, respectively. Three standard JRC surfaces are obtained by stretching the standard surfaces horizontally in proportion, and

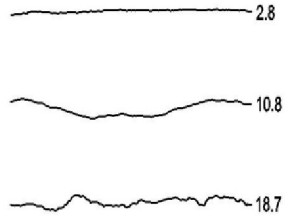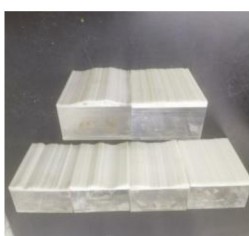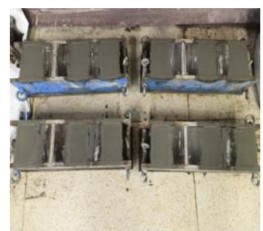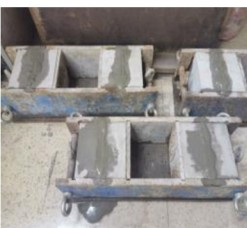

**Fig 2. Specimen production process (a) select JRC curves, (b) Print molds, (c) Pouring specimens, (d) Filling grouting.**

3D models are constructed based on the JRC surfaces, and then three pairs of standard JRC 3D surface molds are constructed by 3D scanning.

Sample preparation uses a steel mold with a standard size of 100mm×100mm×100mm, put the printed mold into the iron mold, pour the proportioned cement mortar into the other side, and then put the steel mold on the shaking table and shake it to ensure the denseness of the poured specimen. After the cement mortar hardened, the samples were dismantled and placed in a standard curing chamber for 21 days at a humidity of 95% and temperature 25 ˚C. Three specimens were prepared for each group to obtain stable test results. Put the maintained joint specimen into the mold, adjust the filling degree and pour the cement paste into the grouting area, vibrate the mold to make the cement paste in close contact with the joint wall. After the cement slurry hardened, the sample was removed and then placed in a standard maintenance room for 14 days, and the specimens were prepared as shown in Fig 2.

## 2.2 Experimental process

YZW-30A microcomputer-controlled electronic rock shear was utilized for uniaxial compression and shearing of the specimen, with a maximum loading capacity of 250KN. The rock sample material and filling material mechanical parameters are shown in Table 1. The shear test system is shown in Fig 3, including the loading system, AE detection system. Shear tests were performed at different normal stresses (2, 3, 4, 5, and 6 MPa respectively) with a shear rate of 0.01 mm/s. Monitoring AE signals during shear to analyze damage evolution. During the test, the normal load was applied to the value set in the test protocol, maintained at a constant level, and subsequently the shear load is applied. During the test, the normal load was applied to the set value of the test scheme remains constant, and then the shear load is applied.

Acoustic emission technology is a dynamic non-destructive testing method used to detect and examine the internal activity of materials under dynamic loading or static conditions. The state of a material can be assessed by capturing, analyzing and utilizing the energy released during its internal damage or fracture of the material. In this experiment, we need to determine the damage evolution of the filled specimen during the shear process and the damage

**Table 1. Mechanical parameters of rock sample materials and filling materials.**

| Materials | Density/kg·m$^{-3}$ | Compressive strength/MPa | Modulus of elasticity/MPa | Cohesive force/MPa | Angle of internal friction/˚ |
|---|---|---|---|---|---|
| Cement mortar | 2291.67 | 47.24 | 1582.61 | 3.61 | 50.69 |
| Cement paste | 1662.56 | 24.61 | 1034.75 | 4.32 | 32.14 |

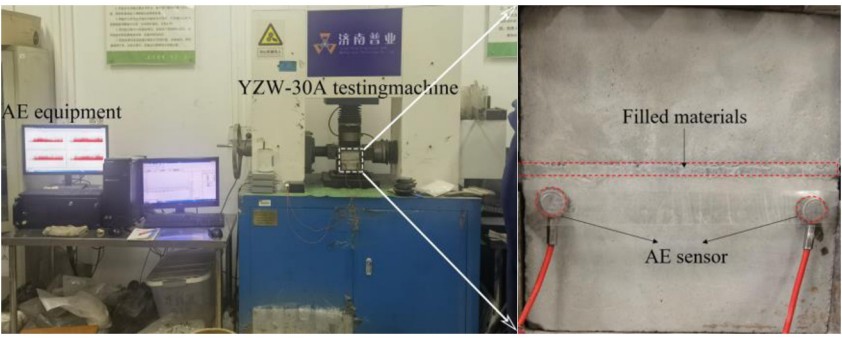

**Fig 3. Experimental systems and AE systems.**

characteristics after the shear is completed, so we choose the acoustic emission technique for characterization.

The 16-channel AE signal monitoring system (PCI-2) was used to record the AE signals of the specimens during the shearing process. Four AE broadband sensors are used to record and localize AE events, as well as count, energy and amplitude information. The four sensors were arranged in the same plane 1 cm from the edge on both the front and back sides of the specimen, while the specimen was modeled, parameters were adjusted and sensor positioning was set up in the acoustic emission system. Achieve planar positioning and monitoring of acoustic emission events. The sensor is fixed on the sample with transparent tape, and a thin layer of petroleum jelly is applied between the sample and the sensor to provide good acoustic coupling. The amplification of the preamplifier and the threshold of the system are both 60 dB during the detection process. To ensure that the shear process is synchronized with the AE acquisition process, the AE acquisition system is triggered at the same time when the shear stress is applied.

## 3 Experimental results

### 3.1 Shear stress-shear displacement curve

The peak shear strengths at different factors obtained based on the filled joint shear test are shown in Table 2. The shear stress curves of the specimens under varying normal stresses are shown in Fig 4. It can be observed that the peak shear strength increases with the increase of normal stress and decreases with the increase of filling degree, and the curves show different forms under different roughness. Before the peak shear strength, with the increase of shear displacement, the shear stress curve of the joints in the shear test are rising, indicating that there is damage in the joints from the beginning of shear, when the main elastic deformation and accompanied by a small amount of plastic deformation. With the accumulation of damage, the plastic deformation increases and the shear strength reaches its peak strength; The residual strength of the joints after shear damage is lower than before, and the damage mechanism is shear-brittle damage. For the post-peak stage, the stress curve shows smoothness and insignificant stress weakening as the shear displacement increases, and the shear stress steadily decreases to the ultimate shear strength.

When the roughness is 10.8 and the normal stress is 2, 3 and 4 MPa, the stress curves all have obvious peak points. When the roughness is 18.7, there are obvious peak points at

**Table 2. Peak shear peak at different factors.**

| JRC | t/d | Strength/MPa | | | | |
|---|---|---|---|---|---|---|
| | | Normal stress/MPa | Normal stress/MPa | Normal stress/MPa | Normal stress/MPa | Normal stress/MPa |
| | | 2 | 3 | 4 | 5 | 6 |
| 2.8 | 0.5 | 2.563 | 4.064 | 4.639 | 5.339 | 5.996 |
| 2.8 | 1.0 | 2.485 | 3.624 | 4.507 | 5.216 | 5.871 |
| 2.8 | 1.5 | 2.259 | 3.574 | 4.256 | 5.007 | 5.708 |
| 10.8 | 0.5 | 2.707 | 4.146 | 5.027 | 5.656 | 6.104 |
| 10.8 | 1.0 | 2.734 | 3.972 | 4.736 | 5.268 | 6.224 |
| 10.8 | 1.5 | 2.366 | 3.851 | 4.541 | 5.151 | 5.941 |
| 18.7 | 0.5 | 3.041 | 4.453 | 5.331 | 6.275 | 6.762 |
| 18.7 | 1.0 | 2.946 | 4.219 | 5.069 | 5.896 | 6.436 |
| 18.7 | 1.5 | 2.705 | 3.925 | 4.826 | 5.504 | 6.186 |

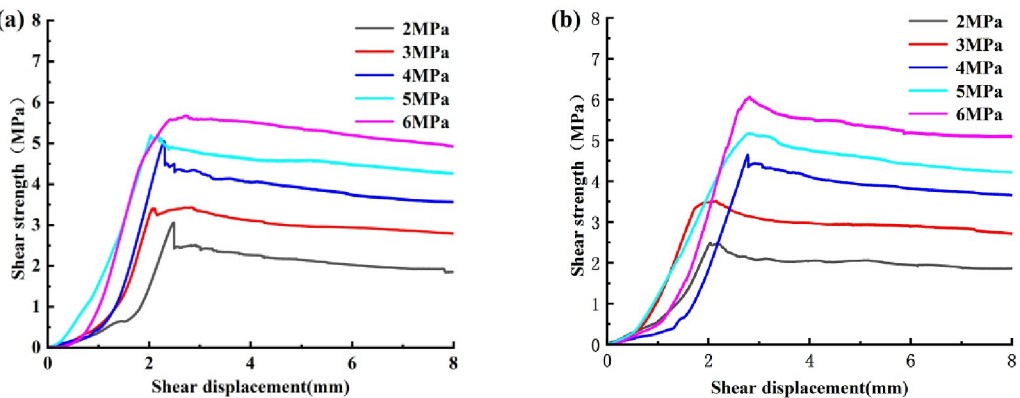

**Fig 4. Stress-strain curves of different JRC joint specimens with a filling degree of 1 (a) JRC = 10.8, (b) JRC = 18.7.**

different normal stresses, and there are large fluctuations in shear stress after the peak with the gradual reduction of shear displacement to the ultimate shear strength. The fluctuation in the process is caused by the local damage of the filling body and the local fracture and concentrated damage of the irregular joint surface. When the filling degree is 1 and 1.5, the shear stress at low and medium normal stress reaches the peak value and then drops suddenly. In contrast, at a filling degree of 0.5, this decreasing change occurs only at a roughness of 10.8. This phenomenon is mainly due to the shear action causing fine damage inside the filled joints. As the shear action continues, the damage within the filled joints, especially in the filled material, accumulates until it reaches the ultimate strength. At medium to high filling degree, the shear force of the raised body of the nodal surface acts on the filling material, which in turn causes the bonding between the nodal surface and the filling material to be broken or the filling material to be destroyed. Under low to medium normal stress, shear plays a dominant role, and after bonding damage, brittle damage occurs under low normal stress with low friction between the nodal surface and the filling material.

During part of the shear experiment, a smaller thumping sound occurs in the specimen, which occurs just as the sudden drop in the shear image indicates that a large amount of energy is being released at this time. In addition, when the filling degree is 1 or 1.5, the post-peak shear stress of the filling joint has a large fluctuation under the medium and low normal stress. It is mainly caused by the shear action which leads to the failure of the joint surface bulge and the fluctuation caused by the fracture.

Existing literature shows that there are two critical filling degrees 0.1 and 1.0, when the filling degree is less than 1.0, the shear strength increases rapidly with the increase of the filling degree, between 0.1 and 1.0, the shear strength decreases with the increase of the filling degree, and when it is greater than 1.0, the shear mechanical parameters are almost constant. Unlike this paper, the shear strength kept decreasing and did not remain almost constant when the filling degree increased from 0.5 to 1.5.

## 3.2 Shear strength characteristics

Fig 5 shows the peak shear strength of each normal stress for three filling degrees, and the peak shear strength of cement slurry filled joints is well described by the linear model proposed by Patton [37] as follows:

$$\tau_p = \sigma_n \tan\theta + c \tag{1}$$

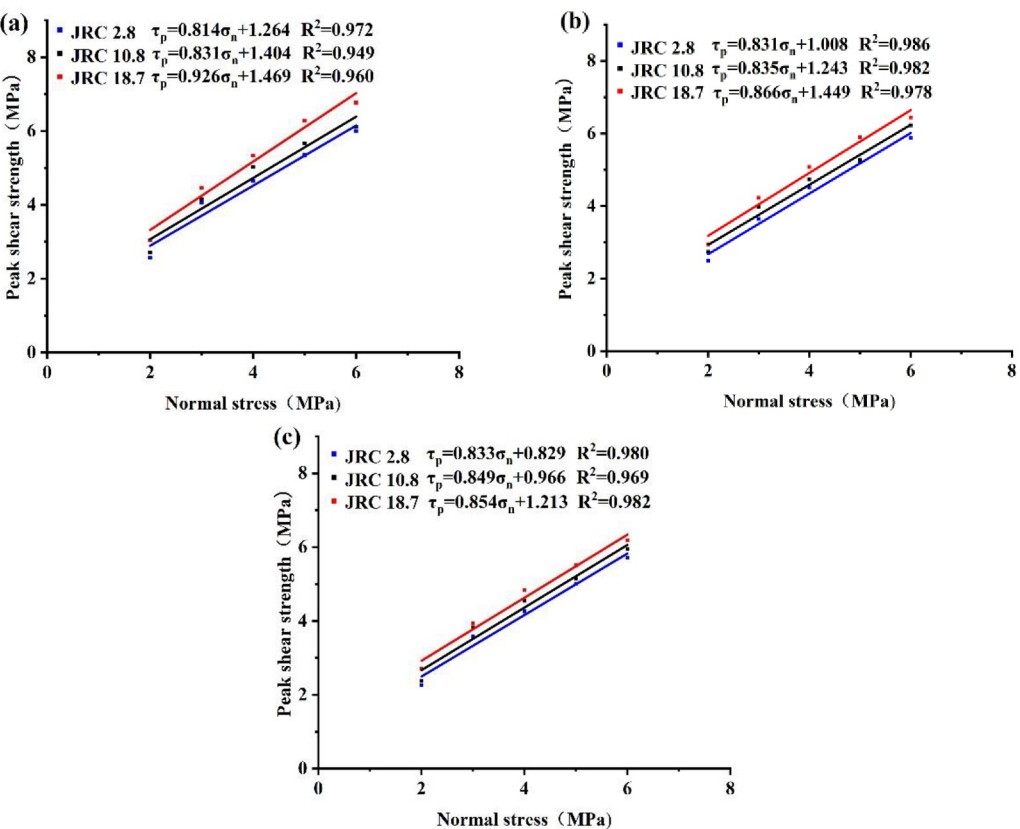

**Fig 5. Variation of peak shear strength with normal stress (a) t/d = 0.5, (b) t/d = 1, (c) t/d = 1.5.**

where $\tau_p$, $\sigma_n$, $\theta$ and $c$ are the peak shear strength, normal stress, friction angle and apparent cohesion, respectively.

Fig 5 shows the variation of peak shear strength with normal stress for different filling degrees. It can be seen that both internal friction angle and cohesion increase with the increase of roughness. There are differences in strength at different roughness, and the differences increase with the increase of normal stress. It shows that the peak shear strength at low normal stress is mainly influenced by the roughness at medium to high roughness. Under medium to high normal stress, the peak shear strength increases due to the compression closure effect, when the normal stress plays a dominant role and amplifies the effect of roughness. Comparison with Fig 5 shows that both the angle of internal friction angle and cohesion decrease with the increase of filling degree at a roughness of 18.7. It shows that due to the increase of the filling degree, the force between the filling material and the joint surface in the shear process decreases, and the shear strength is affected by the cohesive force of the filling material itself. And the difference of internal friction angle between different filling degree roughness decreases, and is gradually similar. It shows that the tendency of decreasing the force at the cementation caused by the increase of filling degree is decreasing, and the tendency of increasing the force affected by the cohesion of filling material itself is increasing. When the filling degree increases, the peak shear strength difference of the three roughness decreases with the increase of normal stress. It shows that with the increase of filling degree, the role of normal stress to increase the peak shear strength is gradually weakened, and the role of normal stress

to enhance the effect of roughness is gradually weakened. At this time, the influence of filling degree on the peak shear strength plays a dominant role.

Analysis of Fig 5 shows that the peak shear strength is positively correlated with the roughness and negatively correlated with the degree of filling. The increase in peak shear strength for roughness of 18.7 relative to 10.8 is greater than the increase in peak shear strength for roughness of 10.8 relative to 2.8. The reason for this is that roughness of 18.7 has larger roughness angles and sharp bumps, and these sharp bumps can still make a greater contribution to the combined shear strength at the three filling degrees. The shear strength mainly depends on the adhesive force between the filling material and the joint surface.

### 3.3 Shear failure mode

Filled joint specimens produce different forms of shear damage, as shown in Fig 6 for the three main failure modes of filled joint:

1. Breakdown of the adhesive surface between the filled joint surface and the filling material (Fig 6(a)). When the roughness of the joint surface is low, the filling material breaks with one of the bonding surfaces of the joint specimen, and the shear strength of the joint depends on the magnitude of the bonding force of the filling material. When the roughness and filling degree are moderate, both bonding surfaces of the filling material and the joint specimen are destroyed, and the shear strength of the joint is affected by the bonding force of the filling material and the frictional resistance of the upper and lower joint projections. It shows that the larger convex body with a roughness of 10.8 prevents the joint from breaking along one bonding surface and causes the failure of both bonding surfaces.

2. Severe damage to filling material and bonding surfaces (Fig 6(b)). When the roughness is 18.7, the damage of the filling material under high normal stress is small. At low normal stresses, the damage of the filling material is larger, and a small number of cracks are generated on the nodal surface. It shows that under high normal stress, the joint surface is excessively compressed and closed, and the degree of joint surface fluctuation is low. At this time, the normal stress plays a leading role, and the filling material is not easy to be destroyed. Under low normal stress, the roughness plays a leading role, and the larger bumps on the joint surface will promote the destruction of the filling material, accompanied by the appearance of joint cracks.

3. The filling material was severely damaged, and the joint surface also showed wear rupture, with more fractures generated and even penetration fractures (Fig 6(c)). Under high

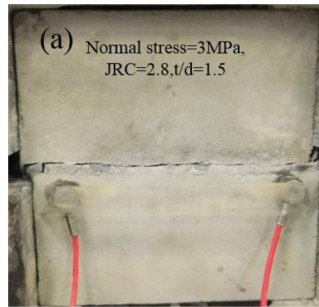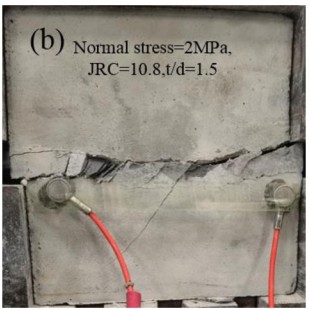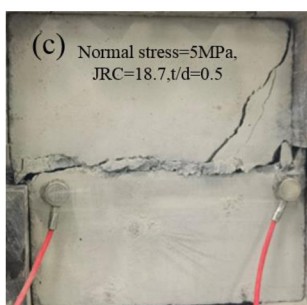

**Fig 6. Shear failure mode of filled joints (a) damage to the adhesive surface of the nodal filler, (b) damage to the adhesive surface of the nodal filler and (c) severe damage to joints and filling material.**

roughness, the increase of normal stress plays a role in aggravating the degree of joint failure, which is contrary to the influence trend of normal stress under medium roughness. This difference shows that when the joint surface fluctuates greatly, the normal stress is high and the filling degree is low, the joint area increases and the bonding effect of the filling material on the joint surface is enhanced. At this time, the failure surface is not continuous and exhibits multiple portions that act from within the filling material back to the bonded surface and then to the joints, so that the filling material and joint damage is intensified, and ultimately there is massive damage. This effect is weakened at low normal stresses and high filling degree.

Based on the experimental results, when the roughness is 2.8, the damage pattern of the joints at different filling degrees is the damage of the joints surface and the adhesive surface of the filling material. A small amount of damage of the filling material may occur at high normal stress, indicating that the filling degree has basically no effect on the shear damage morphology of the filled joints at this roughness, and only the high normal stress causes the difference in damage morphology. When the roughness is 10.8, with the increase of filling degree, the failure of joints evolves from a small amount of joint failure to the failure of filling material, and the failure morphology of filled joints will change with the different filling degree. As the normal stress increases, the degree of damage of the filling material decreases. When the roughness is 18.7, self-destruction of the filling material and fracture of the joints occur. Both filling degree and normal stress have influence on the damage pattern of filled joints, and the increase of normal stress will aggravate the damage of filled joints. However, the increase of filling degree will weaken the effect of normal stress, so that the filled joints damage degree occurs to reduce.

## 4 Damage evolution characteristics of the filled joints

### 4.1 Evolution characteristics of damage degree

The shear stress-shear displacement curve of the filled joint can be divided into four stages: Pre-peak linear stage, pre-peak nonlinear stage, post-peak stage and residual strength stage [37, 38]. Fig 7 shows the evolution pattern of AE events during the shearing process under different test conditions. In stage I, the number of acoustic emission events is very small, and there is no obvious rupture in the nodal specimen. It means that the specimen is elastically deformed in stage I, and weak damage occurs. As the shear displacement increases, cracks continue to sprout and expand along the joint surface, AE events increase rapidly, and the increased shear stress causes the joint to enter a stable fracture extension stage (stage II). The number of AE events remains high in stage I The number of acoustic emission events remained high in phase III after the peak, with a decrease occurring in the middle and late phases. As the shear continues, the main projection and filling body of the nodule undergoes shear damage, and the filled nodule specimen enters into unstable damage. Different damage patterns occur due to shearing under different factors. The number of AE events in stage IV is low. At this time, after the shear failure of most convex bodies and fillers, the specimen slides mainly along the nodal surface and its damage returns to the stable frictional slip stage.

Fig 7(a) and 7(b) shows that the number of acoustic emission events generally tends to increase with the increase of normal stress. It indicates that the increase of normal stress will aggravate the damage of the filled joint specimen. Comparison of Fig 7(c) and 7(d) shows that the relationship between the total number of AE events for the three roughness is 18.7 > 10.8>2.8. The distribution of the number of events is approximately the same for the three roughness in stages I and II, while in stages III and IV, the total number of AE events

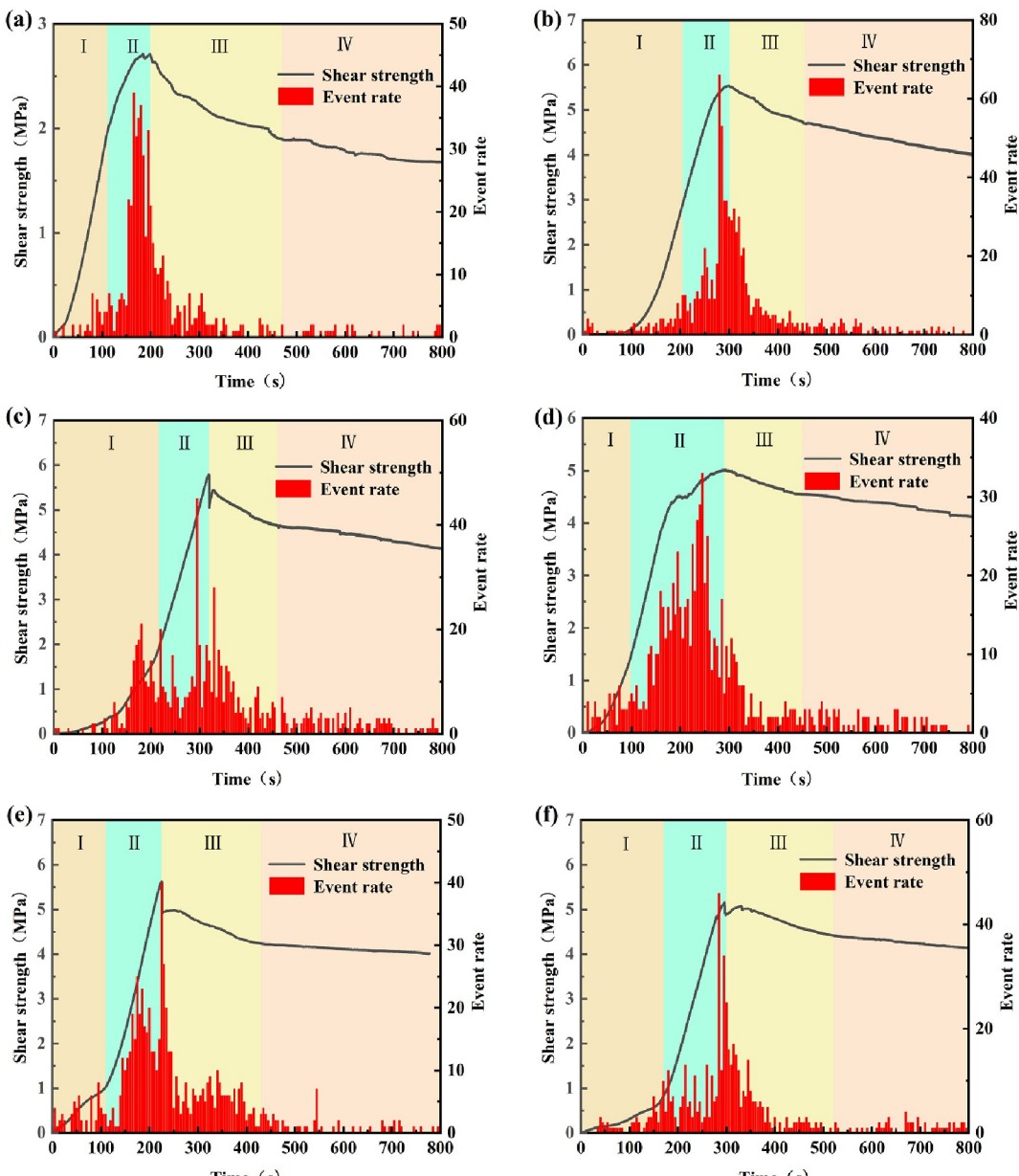

**Fig 7. Damage evolution characteristics of AE events (a) Normal stress 2MPa, (b) Normal stress 5MPa, (c) JRC = 2.8, (d) JRC = 10.8, (e) t/d = 0.5, (f) t/d = 1.**

also shows a relationship of 18.7>10.8>2.8. As the shear damage of joints with high roughness is more serious, the degree of concavity is greater. In the shearing process, the damage of the raised body and the filled body of the joint surface mainly occurs in stages III and IV. Therefore, the number of events is higher during these two stages. Comparison of Fig 7(e) and 7(f) reveals that the relationship between the total number of AE events under the three filling degrees is: 0.5 > 1 > 1.5, and the distribution of AE events in the four stages is approximately the same.

Compared with the failure mode, it is found that when the number of events in the second and third stages is large and evenly distributed, the failure mode of self-destruction of filling material and cracks in joints occurs. A higher number of events corresponds to more severe damage. When the number of events in stage II is the highest, the joint damage of filling material and joints occurs, while when the number of events in the first three stages is high, the damage of the adhesive surface between the joints surface and filling material occurs.

## 4.2 Damage evolution characteristics of AE events

The planar evolution of the AE events in the four stages is shown in Fig 8. The overall evolution of the events is characterized by the occurrence of events in localized areas of the nodal surface, followed by events in most areas of the nodal surface, then events in essentially the entire nodal surface, and finally the continuation of events near the location of the previous events. It indicates that the joint surface is first damaged in a localized area, then the damage occurs near the bonding surface of the joint surface and the filling material and the main projection, finally extends to the damage of the whole joint surface.

In stage I, the distribution of events at low roughness is more uniform, for the greater the filling degree the more uniform distribution can be reflected. In contrast, at medium to high roughness, the event points are clearly found where the degree of concavity of the nodal surface is large. The distribution of event points is more uniform at low roughness in stage II, and the greater the roughness, the more event points in places with high density. It shows that the undulation of the nodal surface is small and the force between the nodal surfaces is small when the roughness is low during the shearing process; Under the condition of medium and high roughness, the position with large concave-convex degree has greater interaction force, which is more prone to joint damage. When the filling degree is greater, the interaction force between the convex bodies on the joint surface decreases, and the degree of joint damage decreases. The much higher distribution density in stage III indicates that large damage to the roughness filled nodules continues in the post-peak stage. Major bumps or fillers facing the shear direction continue to show shear damage, and the filled nodule specimens deform in unstable damage. Stage IV events occur near the event points of the first three stages, because most of the projections and fillers undergo shear damage, their motion is mainly sliding along the joint surface, and the filled nodal specimens enter a stable fracture extension stage. Under medium to high normal stresses, the distribution of event points will occupy the entire nodal surface and behave more densely; Under the condition of low normal stress, the distribution of incident points does not cover the whole joint surface but becomes sparser. In some places, dense normal stress plays a role in aggravating the damage degree, which is consistent with the described joint failure mode.

## 4.3 Energy evolution characteristics

The results of AE energy rate and cumulative energy show that the energy rate and cumulative energy diagram can also be divided into four corresponding stages according to the stage division of shear stress-shear displacement. As shown in Fig 9, the pre-peak linear stage shear stress is in the linear growth stage, which is low and therefore the AE energy rate in this stage is low. The shear stress rises rapidly in the nonlinear stage before the peak, and the AE energy rate shows a nonlinear growth trend as the shear stress increases. The AE energy rate reaches a maximum near the peak shear stress, and the energy rate has a certain delay compared to the appearance of the AE event. In the post-peak stage, the shear stress decreases due to the damage of joint surface and filling material, but the AE energy rate is higher in this stage, especially at the beginning of this stage. In the residual strength stage, the shear stress changes less and

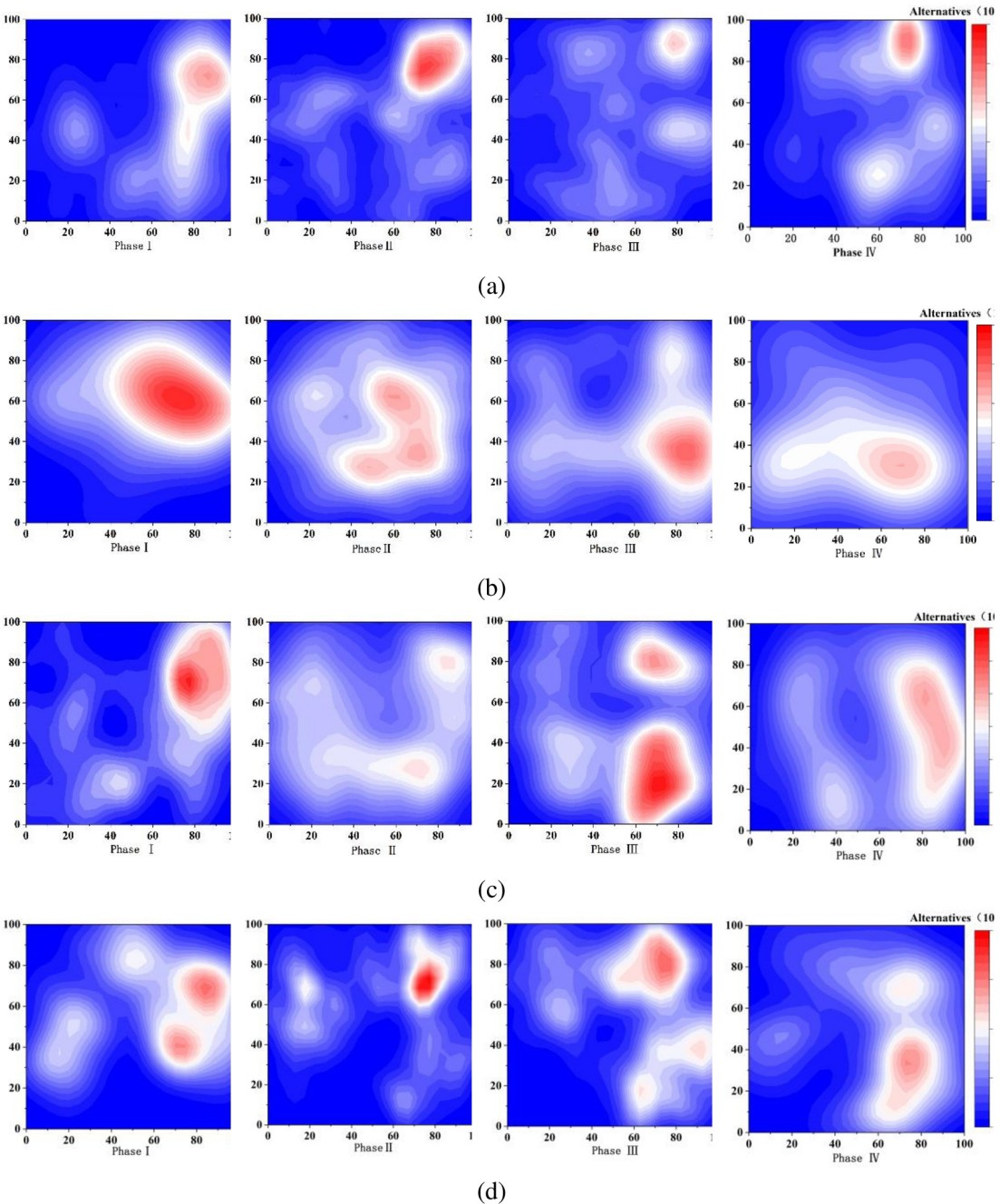

**Fig 8. Diagram of the evolution of AE events (a) JRC = 10.8, t/d = 1, (b) JRC = 2.8, t/d = 1, (c) JRC = 18.7, t/d = 0.5.**

the AE energy rate in this stage is less variable at lower levels. However, under high normal stress and high roughness, there will be a sudden increase in energy rate caused by subsequent shear failure of filling material and wear of joint surface. The effects of normal stress, roughness and filled on the evolution of AE energy are approximately the same as those of the evolution of AE events.

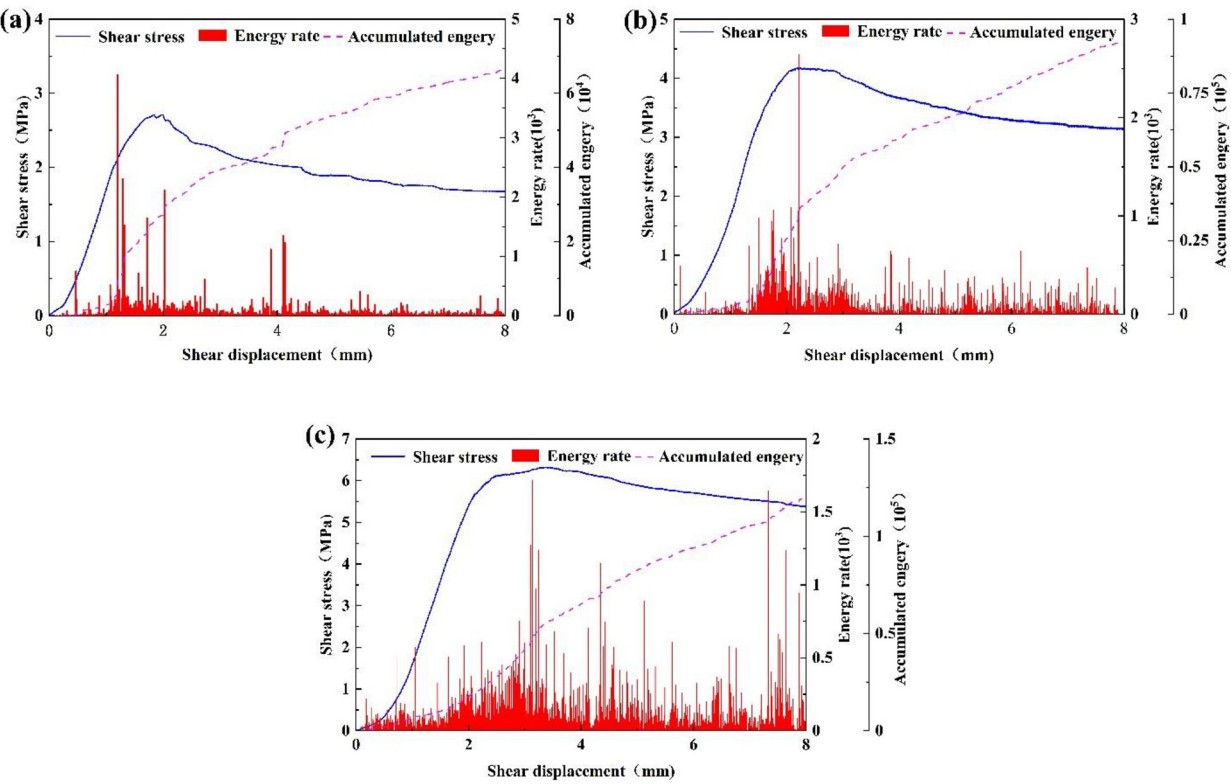

**Fig 9. AE energy rate and cumulative energy (a) normal stress 2MPa, (b) normal stress 4MPa, (c) normal stress 6MPa.**

## 5 Conclusions

1. Normal stress, roughness and filling degree combine to influence the damage pattern of filled joints. At low roughness, it is mainly the damage of the adhesive surface between the filling material and the joint surface. With the increase of the normal stress, the damage pattern is from the weak damage of the joint to the damage of the filling material to the damage of both the filling material and the joint surface, and the normal stress will aggravate the damage degree of the joint.

2. Under the low and medium normal stress, the shear stress of the filled joints at low filling degree undergoes a sudden drop after reaching the peak, indicating that brittle damage occurs after bonding damage and low normal stress with low friction between the joint surface and the filling material. There are large fluctuations in the post-peak shear stress of the filled joints at medium and high filling degrees, mainly due to the fluctuations caused by the shear action leading to the destruction of the raised body of the joint surface and the fracture generation.

3. The peak shear strength of the filled joints is positively correlated with the roughness and negatively correlated with the filling degree. With the increase of filling degree, the role of normal stress to increase the peak shear strength gradually decreases, and the role of normal stress to enhance the effect of roughness gradually decreases. Under the same normal stress, with the increase of filling degree, the main influence of peak shear strength gradually changed from roughness to filling degree.

4. The damage degree of filled joints during shear is positively correlated with normal stress and roughness, and negatively correlated with filling degree. In the shear-elastic deformation stage, the filled joints have almost no damage; near the peak shear stress. The filled joints mainly undergo damage of larger protrusions and filling material, and the AE events increase sharply; the filled joints enter the stable frictional slip stage in the post-peak residual stage, and the number of events decrease.

## Author Contributions

**Conceptualization:** Zhilin Shu.

**Funding acquisition:** Yang Liu, Binyu Luo.

**Methodology:** Yang Liu, Yicheng Ye, Binyu Luo.

**Supervision:** Weiqi Wang.

**Writing – original draft:** Zhilin Shu.

**Writing – review & editing:** Weiqi Wang, Jinpeng Jia.

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
