## [Decision Letter · Decision Letter 0]

24 Jan 2024

PONE-D-23-44098Influences of roughness and filling degree on the shear strength and damage evolution characteristics of cement-filled jointsPLOS ONE

Dear Dr. Liu,

Thank you for submitting your manuscript to PLOS ONE. After careful consideration, we feel that it has merit but does not fully meet PLOS ONE’s publication criteria as it currently stands. Therefore, we invite you to submit a revised version of the manuscript that addresses the points raised during the review process.

We look forward to receiving your revised manuscript.

Kind regards,

Amirsalar Khandan, Ph.D.

Academic Editor

PLOS ONE

Journal Requirements:

"The Wuhan Knowledge Innovation Special Program (2022010801020307), The 2022 Hubei Emergency Management Department Special Funds for Safety Production Program (SJZX20220907), Hubei Provincial Natural Science Foundation Youth Program(2022CFB590)."               

"The authors declare no conflict of interest. The funders had no role in the design of the study; in the collection, analyses, or interpretation of data; in the writing of the manuscript; or in the decision to publish the results."

5. PLOS requires an ORCID iD for the corresponding author in Editorial Manager on papers submitted after December 6th, 2016. Please ensure that you have an ORCID iD and that it is validated in Editorial Manager. To do this, go to ‘Update my Information’ (in the upper left-hand corner of the main menu), and click on the Fetch/Validate link next to the ORCID field. This will take you to the ORCID site and allow you to create a new iD or authenticate a pre-existing iD in Editorial Manager. Please see the following video for instructions on linking an ORCID iD to your Editorial Manager account: https://www.youtube.com/watch?v=_xcclfuvtxQ.

6. Please amend the manuscript submission data (via Edit Submission) to include author "Weiqi Wang".

Reviewers' comments:

Reviewer's Responses to Questions

**Comments to the Author**

1. Is the manuscript technically sound, and do the data support the conclusions?

Reviewer #1: Yes

Reviewer #2: Yes

2. Has the statistical analysis been performed appropriately and rigorously? 

Reviewer #1: Yes

Reviewer #2: Yes

3. Have the authors made all data underlying the findings in their manuscript fully available?

Reviewer #1: Yes

Reviewer #2: No

4. Is the manuscript presented in an intelligible fashion and written in standard English?

Reviewer #1: Yes

Reviewer #2: Yes

5. Review Comments to the Author

Reviewer #1: The paper titled "Influences of roughness and filling degree on the shear strength and damage evolution characteristics of cement-filled joints" presents a comprehensive study on the shear resistance of filled rock joints, focusing on the effects of roughness and filling degree. The research is situated in the context of rock mechanics and geotechnical engineering, where understanding the behavior of rock joints, especially when filled with materials like cement, is crucial for various engineering applications.

Overall, this paper is well-organized and well-prepared. However, there are several issues should be addressed. Please see my comments as follows:

1. The paper requires a thorough language check for grammatical correctness and clarity as many typos and grammar issues can be found in the manuscript.

2. The literature review appears to be somewhat limited in scope. Including more recent and diverse studies could provide a more comprehensive background and better context for your research. The following papers for your reference: 10.1080/19648189.2023.2172083; 10.1007/s00603-023-03486-x; 10.1002/nag.3002; 10.1007/s00603-023-03690-9; 10.1007/s00603-023-03432-x

3. The explanation of DIC and AE techniques require further clarification. More detailed procedural descriptions would enhance the replicability of the study.

4. Consider adding a subsection on the rationale behind choosing these specific methods.

5. Comparing and contrasting your results with existing literature would add depth to the discussion. It would benefit from a clearer articulation of the practical implications of your findings, especially how they can be applied in real-world scenarios.

Overall, the study presents interesting and valuable research. In summary, while the manuscript presents important research with potential applications in the field of rock mechanics, addressing the above suggestions would significantly enhance its overall quality and suitability for publication.

Reviewer #2: The paper evaluates the effects of normal stress, roughness and filling degree on the shear strength, failure mode and damage evolution of cement-filled joints are studied by combining with acoustic emission (AE) technology, which provides the experimental verification for the effect of cement filling in reinforcing rock joints. Accordingly, I would recommend the manuscript needs major revision before it is considered for publication. Please, see comments below.

1. The first sentence of the abstract is overstated.

2. “Test results with different normal stress, roughness and filling degree” is not clear.

3. The shear mechanical properties in this paper remain limited to the two-dimensional stage. Currently, many researches have been carried out on the shear properties of rock joint under different roughness and normal stress. What are the innovations?

4. Define “critical filling degree”.

5. The introduction of rock-like materials should be deleted, and the research status of shear damage evolution characteristics of filled joints should be added.

6. The filling thickness and maximum undulation height of the joints under the three filling degrees is not clear. Please, more details.

7. Please provide the basic mechanical parameters of rock-like materials and filling materials

8. How to verify the similarity of surface morphology of joint samples?

9. The font size and scale orientation of all pictures are inconsistent.

10. “When the roughness and normal stress are not large”, “when the roughness is moderate.” These descriptions are not accurate. The description of the most experimental laws is difficult for readers to understand.

11. The test conditions in Fig. 6 need to be labeled.

12. In the third chapter, most of the test result analysis does not present figures or data support.

13. “It shows that with the increase of filling degree, the effect of normal stress increasing peak shear strength gradually decreases, and the effect of normal stress enhancing the effect of roughness gradually decreases.” is not clear. Much more detailed explanation needed.

14. Please add the diagram of the evolution of AE events in the same roughness and different filling degrees.

6. PLOS authors have the option to publish the peer review history of their article (what does this mean?). If published, this will include your full peer review and any attached files.

Reviewer #1: No

Reviewer #2: No

---

## [Author Response · Author response to Decision Letter 0]

16 Mar 2024

Response to the review comments 

Journal: PLOS ONE

Manuscript Number: PONE-D-23-44098

Title: Influences of roughness and filling degree on the shear strength and damage evolution characteristics of cement-filled joints

Author(s): Zhilin Shu; Yang Liu; Yicheng Ye; Weiqi Wang; Binyu Luo; Jinpeng Jia

Dear Editor-in-Amirsalar Khandan, Ph.D. and Reviewers:

Thank you for your letter and for the reviewers’ comments concerning our manuscript entitled “Influences of roughness and filling degree on the shear strength and damage evolution characteristics of cement-filled joints”. Those comments are all very valuable and helpful for revising and improving our manuscript, as well as the important guiding significance to our further research. We have studied the comments carefully and have corrected them. We hope it will meet with your approval. The revised portions are marked in red in the manuscript. The main corrections in the manuscript and the responses to the reviewer’s comments are as flowing.

We appreciate editors-in- Amirsalar Khandan, Ph.D and the Reviewers’ warm work earnestly and hope that the correction will meet with approval.

Finally, thank you again sincerely for your valuable advice.

Best regards.

Yang Liu on behalf of all authors

Corresponding author:

Name: Yang Liu

Telephone: +86 135-9422-8774

E-mail: liuyang@wust.edu.cn

School of Resources and Environmental Engineering, 

Wuhan University of Science and Technology, 

Wuhan, 430081, P.R. China

Responds to the reviewer’s comments:

Reviewer #1

Response: Thank you for your recognition of our research, which is an important inspiration for our future research. Your pertinent and constructive comments are very helpful, and we have revised and improved the manuscript based on your comments.

1. The paper requires a thorough language check for grammatical correctness and clarity as many typos and grammar issues can be found in the manuscript.

Response: Thank you for your good suggestion. We have thoroughly revised the paper for grammar and clarity. Specific changes have been highlighted in red in the text.

2. The literature review appears to be somewhat limited in scope. Including more recent and diverse studies could provide a more comprehensive background and better context for your research.

Response: Thanks for your valuable suggestion. In order to give the study a more comprehensive context, we include recent and diverse studies in the introduction (Section 1, paragraph 2 and paragraph 4).

Revision: The ESFC differs from the composite rock mass mentioned earlier, thus the expansion and extrusion of the expansive slurry cause notable variations in the mechanical properties of the composite rock mass compared to ordinary composite rock mass. To investigate the effect of volume expansion on the shear properties of the composite rock mass, firstly, the ESFC models were established using the PFC2D to simulate the volume expansion behavior of the expansive slurry. Subsequently, direct shear tests were conducted on both the OSFC and the ESFC under the same conditions of the laboratory tests. The shear mechanical characteristics, damage evolution, contact force distribution characteristics and particle displacement of the composite rock mass with internal expansion stress were analyzed, and the reinforcement mechanism of the expansive slurry was discussed from a mesoscopic perspective.

3. The explanation of DIC and AE techniques require further clarification. More detailed procedural descriptions would enhance the replicability of the study.

Response: Thank you for your excellent suggestion. This study only uses AE and does not use the DIC, so there is no introduction to the DIC. we have also added a more detailed program description of AE (Section 2.2, paragraph 3).

Revision: The 16-channel AE signal monitoring system (PCI-2) was used to record the AE signals of the specimens during the shearing process. Four AE broadband sensors are used to record and localize AE events, as well as count, energy and amplitude information. The four sensors were arranged in the same plane 1 cm from the edge on both the front and back sides of the specimen, while the specimen was modeled, parameters were adjusted and sensor positioning was set up in the acoustic emission system. Realize the plane positioning and monitoring of acoustic emission events. The sensor is fixed on the sample with transparent tape, and a thin layer of petroleum jelly is applied between the sample and the sensor to provide good acoustic coupling. The amplification of the preamplifier and the threshold of the system are both 60 dB during the detection process. To ensure that the shear process is synchronized with the AE acquisition process, the AE acquisition system is triggered at the same time when the shear stress is applied.

4. Consider adding a subsection on the rationale behind choosing these specific methods.

Response: Thanks for your good suggestion. We have added a subsection to justify the choice of the AE characterization method as you suggested (Section 2.2, paragraph 2).

Revision: Acoustic emission technology is a dynamic non-destructive testing method used to detect and examine the internal activity of materials under dynamic loading or static conditions. The state of a material can be assessed by capturing, analyzing and utilizing the energy released during internal damage or fracture of the material. In this experiment, we need to determine the damage evolution of the filled specimen during the shear process and the damage characteristics after the shear is completed, so we choose the acoustic emission technique for characterization.

5. Comparing and contrasting your results with existing literature would add depth to the discussion. It would benefit from a clearer articulation of the practical implications of your findings, especially how they can be applied in real-world scenarios.

Response: Thank you for your good suggestion. We added a comparison with the existing literature in section 3.1, paragraph 4 of the article.

Revision: Comparing and contrasting your results with existing literature would add depth to the discussion. It would benefit from a clearer articulation of the practical implications of your findings, especially how they can be applied in real-world scenarios.

Reviewer 2#

Thanks for your comments of this manuscript, your pertinent and constructive comments are an important inspiration for our research, and we have revised and improved the manuscript based on your comments.

1. The first sentence of the abstract is overstated.

Response: Thank you for your good advice. We have changed the first sentence of the abstract.

Revision: The shear resistance of filling joints is an important factor affecting the stability of rock joints.

2. “Test results with different normal stress, roughness and filling degree” is not clear.

Response: Thank you for your excellent suggestion. Currently, we have added the basic information about expansive slurry to the manuscript, the specific details have been presented in Section 2.1.

Revision: The uniaxial compression and shear tests on cement pastes with a water-cement ratio of 0.7 and an expansion agent content of 0% (normal slurry) and 10% (expansive slurry) have been carried out, the measured mechanical parameters are shown in Table 1.

Table1 Material parameters

Parameter Density/kg.m-3 UCS/MPa Elastic modulus/MPa Shear strength/MPa

Rock 2078.77 47.47 1683.33 6.15

Ordinary slurry 1682.80 26.28 1074.75 6.28

Expansive slurry 1721.80 25.73 1170.09 6.26

3. The shear mechanical properties in this paper remain limited to the two-dimensional stage. Currently, many researches have been carried out on the shear properties of rock joint under different roughness and normal stress. What are the innovations?

Response: Thank you for your excellent question. Firstly, we started the study on the basis of filling the joints. Secondly, I asked to explore the multifactorial combination of factors affecting the shear characteristics of the filled joints. Moreover, we combined the acoustic emission characterization means to investigate the damage evolution characteristics of the joint surface.

4. Define “critical filling degree”.

Response: Thank you for your good suggestion. The critical filling degree is when the filling degree is between the two sides of the critical filling degree, the law of shear mechanical properties will change. We have included a more detailed description in the third paragraph of the introduction.

Revision: When the filling degree is less than 1.0, the shear strength increases rapidly with increasing degree of filling. Between 0.1 and 1.0, the shear strength decreases with increasing filling degree. At greater than 1.0, the shear mechanical parameters are almost constant.

5. The introduction of rock-like materials should be deleted, and the research status of shear damage evolution characteristics of filled joints should be added.

Response: Thank you for your excellent suggestion. We follow the suggestion to delete the introduction of rock-like materials and add the research status of shear damage evolution characteristics of filled joints (Section 1, paragraph 4).

Revision: Acoustic emission (AE) technology enables the characterization of damage evolution in specimens through microscopic monitoring. AE technology is simple to operate and requires low samples. The evolution of AE parameters mainly includes the analysis of events, impacts, counts and energies. Shang monitored the rupture process inside the specimen during shear by the AE technique. The AE events were found to be dispersed in the pre-peak phase but confined around the cleavage surface in the post-peak phase. Ostapchuk studied the acoustic emission behavior of filled fissures in different states of shear deformation. The analysis of the evolution of the acoustic emission parameters allows to construct the process of damage evolution inside the specimen. Meng found that for filled joints, the peak event rate and energy rate are mainly obtained at or near the turn of the shear stress curve. The evolution of acoustic emission with displacement can accurately reflect the shear damage mechanism during shearing.

6. The filling thickness and maximum undulation height of the joints under the three filling degrees is not clear. Please, more details. 

Response: Thank you for your good question. We include more details on the filling thickness and maximum heave difference at the three filling degrees in Section 2.1, paragraph 1 of the article.

Revision: The undulation height differences of the three curves are 1.7 mm, 5.4 mm, and 4.4 mm, respectively. i.e., the three filling degrees correspond to filling thicknesses of 0.85, 1.7, and 2.55 mm, 2.7, 5.4, and 8.1 mm, and 2.2, 4.4, and 6.6 mm, respectively.

7. Please provide the basic mechanical parameters of rock-like materials and filling materials.

Response: Thank you for your good question We provide tables of the basic mechanical parameters of rock-like materials and filling materials in Section 2.2 of this article.

Revision:

Table. 1 Mechanical parameters of rock sample materials and filling materials

Materials Density

Kg/m3 Compressive strength/MPa Modulus of elasticity/MPa Cohesive force/MPa Angle of internal friction/°

Cement mortar

Cement paste 2291.67

1662.56 47.24

24.61 1582.61

1034.75 3.61

4.32 50.69

32.14

8. How to verify the similarity of surface morphology of joint samples?

Response: The specimens are obtained by casting, and both the upper and lower disks of the specimens with the same roughness match each other. Therefore, it can be verified that the nodal surface is similar in shape.

9. The effect of swelling slurry modulus not included in the study.

Response: Thank you for your good question. We have adjusted the font size and scale orientation of the image. The legend for the adjusted image is marked red in the article.

10. “When the roughness and normal stress are not large”, “when the roughness is moderate.” These descriptions are not accurate. The description of the most experimental laws is difficult for readers to understand.

Response: Thank you for your good question. We have corrected the inaccurate descriptions in the article, which have been redlined in the article.

11. The test conditions in Fig. 6 need to be labeled.

Response: Thank you for your good suggestion. We have labeled the test conditions in Figure 6.

Revision: 

Fig. 6 Shear failure mode of filled joints: (a) Damage to the adhesive surface of the nodal filler, (b) Damage to the adhesive surface of the nodal filler and (c) Severe damage to joints and filling material

12. In the third chapter, most of the test result analysis does not present figures or data support.

Response: Thank you for your good suggestion. In response to Chapter 3, most of the test results analyzed are not given graphical or data support. We have added a statistical table of peak shear strength at different factors in Chapter 3.

Revision:

Table. 2 Peak shear strength at different factors

 Stress=2MPa Stress=3MPa Stress=4MPa Stress=5MPa Stress=6MPa

JRC=2.8 t/d=0.5

JRC=2.8 t/d=1.0

JRC=2.8 t/d=1.5

JRC=10.8 t/d=0.5

JRC=10.8 t/d=1.0

JRC=10.8 t/d=1.5

JRC=18.7 t/d=0.5

JRC=18.7 t/d=1.0

JRC=18.7 t/d=1.5 2.563

2.485

2.259

2.707

2.734

2.366

3.041

2.946

2.705 4.064

3.624

3.574

4.146

3.972

3.851

4.453

4.219

3.925 4.639

4.507

4.256

5.027

4.736

4.541

5.331

5.069

4.826 5.339

5.216

5.007

5.656

5.268

5.151

6.275

5.896

5.504 5.996

5.871

5.708

6.104

6.224

5.941

6.762

6.436

6.186

13. “It shows that with the increase of filling degree, the effect of normal stress increasing peak shear strength gradually decreases, and the effect of normal stress enhancing the effect of roughness gradually decreases.” is not clear. Much more detailed explanation needed.

Response: Thank you for your excellent suggestion. We explain this in more detail in section 3.2, paragraph 2.

Revision: Comparison with Fig. 5 shows that both the angle of internal friction angle and cohesion decrease with the increase of filling degree at a roughness of 18.7, It shows that due to the increase of the filling degree, the force between the filling material and the joint surface in the shear process decreases, and the shear strength is affected by the cohesive force of the filling material itself. And the difference of internal friction angle between different filling degree roughness decreases, and is gradually similar. It shows that the tendency of decreasing the force at the cementation caused by the increase of filling degree is decreasing, and the tendency of increasing the force affected by the cohesion of filling material itself is increasing. When the filling degree increases, the peak shear strength difference of the three roughnesses decreases with the increase of normal stress. It shows that with the increase of filling degree, the role of normal stress to increase the peak shear strength is gradually weakened, and the role of normal stress to enhance the effect of roughness is gradually weakened. At this time, the influence of filling degree on the peak shear strength plays a dominant role.

14. Please add the diagram of the evolution of AE events in the same roughness and different filling degrees.

Response: Thank you for your good suggestion. We have added the diagram of the evolution of AE events in the same roughness and different filling degrees.

Revision:

(a)

(b)

(c)

(d)

Fig. 8 Diagram of the evolution of AE events: (a) JRC=10.8 t/d=1, (b) JRC=2.8 t/d=1, (c) JRC=18.7 t/d=0.5 and (d)JRC=18.7 t/d=1

Other changes:

We tried our best to improve the manuscript and made some changes in the manuscript. These changes will not influence the content and framework of the paper. And here we did not list the changes but marked them in red in the revised manuscript.

---

## [Decision Letter · Decision Letter 1]

16 Apr 2024

PONE-D-23-44098R1Influences of roughness and filling degree on the shear strength and damage evolution characteristics of cement-filled jointsPLOS ONE

Dear Dr. Liu,

Thank you for submitting your manuscript to PLOS ONE. After careful consideration, we feel that it has merit but does not fully meet PLOS ONE’s publication criteria as it currently stands. Therefore, we invite you to submit a revised version of the manuscript that addresses the points raised during the review process.

**-What are the two main factors investigated in this study regarding cement-filled joints?**

**-How was the roughness of the joints measured or characterized in the experiments?**

**-What techniques were used to evaluate the damage evolution characteristics of the cement-filled joints under shear loading?**

**-Did the results show any differences in the shear strength and damage evolution between joints with different roughness and filling degrees? If so, what were the key findings?**

**-Language edit needed**

We look forward to receiving your revised manuscript.

Kind regards,

Amirsalar Khandan, Ph.D.

Academic Editor

PLOS ONE

Journal Requirements:

Reviewers' comments:

Reviewer's Responses to Questions

**Comments to the Author**

1. If the authors have adequately addressed your comments raised in a previous round of review and you feel that this manuscript is now acceptable for publication, you may indicate that here to bypass the “Comments to the Author” section, enter your conflict of interest statement in the “Confidential to Editor” section, and submit your "Accept" recommendation.

Reviewer #1: All comments have been addressed

Reviewer #3: All comments have been addressed

2. Is the manuscript technically sound, and do the data support the conclusions?

Reviewer #1: Yes

Reviewer #3: Yes

3. Has the statistical analysis been performed appropriately and rigorously? 

Reviewer #1: Yes

Reviewer #3: Yes

4. Have the authors made all data underlying the findings in their manuscript fully available?

Reviewer #1: Yes

Reviewer #3: Yes

5. Is the manuscript presented in an intelligible fashion and written in standard English?

Reviewer #1: No

Reviewer #3: Yes

6. Review Comments to the Author

**Reviewer #1:** As the authors have addressed all the comments accordingly, the paper can be accepted after check the typos and grammar issues of the whole paper.

**Reviewer #3**: The shear resistance of filling joints in rock formations is crucial for stability, prompting pressure-shear tests on cement-filled joints, complemented by acoustic emission (AE) analysis. Results reveal that increasing roughness leads to more intricate failure modes, with low roughness primarily damaging the bonding interface, while high roughness causes joint failure to escalate. Peak shear strength positively correlates with roughness but negatively with filling degree. Higher filling degrees weaken roughness effects due to filling material, while normal stress amplifies them. AE evolution indicates a positive correlation between damage degree and normal stress/roughness but a negative correlation with filling degree. Damage initiates locally and progresses towards joint failure. This insight aids in assessing and managing the stability of cement-filled rock joints.

Paper Comments:

The following statements are some comments about the paper:

• The introduction should provide a robust emphasis on the research, offering a comprehensive explanation of the entire process, spanning past, present, and future scope. Highlight how the present study enhances accuracy compared to previous research endeavours. Strengthen the introduction by integrating recent advancements in the field and identifying potential research gaps. It's highly recommended to incorporate recent literature to enrich the discussion and provide additional context for the study.

The introduction section can be furnished with some new papers like:

a. https://doi.org/10.1016/j.compgeo.2024.106136

b. https://doi.org/10.1016/j.ijrmms.2024.105721

c. https://doi.org/10.1061/IJGNAI.GMENG-9366

d. About error detection with new method: doi: 10.1109/ICSP54964.2022.9778676.

- Importance and Novelty of the Selected Problem:

• Explain why the problem is significant in its field or industry.

• Highlight any unique aspects or approaches that set it apart.

• Emphasize how addressing this problem advances knowledge or fills a gap in the field.

1. Paper writing method and Quality:

• Highlight your key results and contribution in abstract.

• The motivation of the paper should be improved. And please write your research contribution with number order.

• Write organization of your paper in the end of introduction.

• Please write background and motivation of study clearly in introduction.

• Please check the whole manuscript for types and grammar errors. Language of the paper should be improved.

• Some minor grammatical mistakes are there, read carefully and correct them.

• In the conclusion part please write the exact improvement number by using your proposed method..

2. Figures:

• The text inside the figures are not clear.

• Adjust all figures as it is not well structured.

• Cite each figure in text look which one is not cited.

3. Tables:

• Need little bit adjustment of tables.

• Cite each table in text where it is needed.

4. References:

• Please use the most recent references for the paper, i.e., starting from 2019-2020 till date.

5. Recommendations:

• Authors are requested to make typesetting according to the paper template on the journal website as it is not up to the mark.

• This study has merit for publication. However, I would recommend a minor revision to improve the quality of the manuscript.

7. PLOS authors have the option to publish the peer review history of their article (what does this mean?). If published, this will include your full peer review and any attached files.

Reviewer #1: No

Reviewer #3: No

---

## [Author Response · Author response to Decision Letter 1]

1 Nov 2024

Responds to the reviewer’s comments:

1. What are the two main factors investigated in this study regarding cement-filled joints?

Response: Thank you for your good suggestion. The main factors are roughness, filling and normal stress.

2. How was the roughness of the joints measured or characterized in the experiments?

Response: Thanks for your valuable suggestion. Barton proposed the joint surface roughness standard, using JRC to quantitatively describe the surface roughness of rock joints, and this paper selects three of the ten standard roughness curves defined by it, and the roughness of the poured sample corresponds to the three standard curves.

3. What techniques were used to evaluate the damage evolution characteristics of the cement-filled joints under shear loading?

Response: Thank you for your excellent suggestion. The whole shear process was monitored by acoustic emission technology, and the number of events, event locations and energy were obtained to evaluate the damage evolution characteristics of cement-filled joints under shear load.

4. Did the results show any differences in the shear strength and damage evolution between joints with different roughness and filling degrees? If so, what were the key findings?

Response: Thanks for your good suggestion. There are differences in the shear strength and damage evolution of joints with different roughness and filling degrees, and the specific differences are as follows. The peak shear strength of the filled joint specimen was positively correlated with roughness and negatively correlated with the filling degree. During the shearing process, the damage degree of the filled joint specimen was positively correlated with the roughness, and negatively correlated with the filling degree.

Other changes:

We tried our best to improve the manuscript and made some changes in the manuscript. These changes will not influence the content and framework of the paper. And here we did not list the changes but marked them in red in the revised manuscript.

---

## [Decision Letter · Decision Letter 2]

7 Nov 2024

Influences of roughness and filling degree on the shear strength and damage evolution characteristics of cement-filled joints

PONE-D-23-44098R2

Dear Dr. Liu,

We’re pleased to inform you that your manuscript has been judged scientifically suitable for publication and will be formally accepted for publication once it meets all outstanding technical requirements.

Kind regards,

Jiaolong Ren

Academic Editor

PLOS ONE

Additional Editor Comments (optional):

Reviewers' comments:

Reviewer's Responses to Questions

**Comments to the Author**

1. If the authors have adequately addressed your comments raised in a previous round of review and you feel that this manuscript is now acceptable for publication, you may indicate that here to bypass the “Comments to the Author” section, enter your conflict of interest statement in the “Confidential to Editor” section, and submit your "Accept" recommendation.

Reviewer #1: All comments have been addressed

2. Is the manuscript technically sound, and do the data support the conclusions?

Reviewer #1: Yes

3. Has the statistical analysis been performed appropriately and rigorously? 

Reviewer #1: Yes

4. Have the authors made all data underlying the findings in their manuscript fully available?

Reviewer #1: Yes

5. Is the manuscript presented in an intelligible fashion and written in standard English?

Reviewer #1: Yes

6. Review Comments to the Author

Reviewer #1: As the authors have addressed all my comments, I have no further comments and this manuscript can be published after polish.

7. PLOS authors have the option to publish the peer review history of their article (what does this mean?). If published, this will include your full peer review and any attached files.

Reviewer #1: **Yes: **Changtai Zhou

---

## [Editor Report · Acceptance letter]

9 Jan 2025

PONE-D-23-44098R2 

PLOS ONE

Dear Dr. Liu, 

I'm pleased to inform you that your manuscript has been deemed suitable for publication in PLOS ONE. Congratulations! Your manuscript is now being handed over to our production team.

Kind regards, 

on behalf of

Dr. Jiaolong Ren 

Academic Editor

PLOS ONE